# Dimethyl Fumarate Induces Apoptosis via Inhibition of NF-κB and Enhances the Effect of Paclitaxel and Adriamycin in Human TNBC Cells

**DOI:** 10.3390/ijms23158681

**Published:** 2022-08-04

**Authors:** Katsumasa Tsurushima, Masanobu Tsubaki, Tomoya Takeda, Takuya Matsuda, Akihiro Kimura, Honoka Takefuji, Akane Okada, Chiaki Sakamoto, Toshihiko Ishizaka, Shozo Nishida

**Affiliations:** 1Division of Pharmacotherapy, Faculty of Pharmacy, Kindai University, Kowakae, Higashiosaka 577-8502, Japan; 2Sakai City Medical Center, Department of Pharmacy, Sakai 593-8304, Japan

**Keywords:** adriamycin, dimethyl fumarate, NF-κB, paclitaxel, triple-negative breast cancer

## Abstract

Triple-negative breast cancer (TNBC) has the poorest prognosis of all breast cancer subtypes. Recently, the activation of NF-κB, which is involved in the growth and survival of malignant tumors, has been demonstrated in TNBC, suggesting that NF-κB may serve as a new therapeutic target. In the present study, we examined whether dimethyl fumarate (DMF), an NF-κB inhibitor, induces apoptosis in TNBC cells and enhances the apoptosis-inducing effect of paclitaxel and adriamycin. Cell survival was analyzed by the trypan blue assay and apoptosis assay. Protein detection was examined by immunoblotting. The activation of NF-κB p65 was correlated with poor prognosis in patients with TNBC. DMF induced apoptosis in MDA-MB-231 and BT-549 cells at concentrations that were non-cytotoxic to the normal mammary cell line MCF-10A. Furthermore, DMF inhibited NF-κB nuclear translocation and Survivin, XIAP, Bcl-xL, and Bcl-2 expression in MDA-MB-231 and BT-549 cells. Moreover, DMF enhanced the apoptosis-inducing effect of paclitaxel and adriamycin in MDA-MB-231 cells. These findings suggest that DMF may be an effective therapeutic agent for the treatment of TNBC, in which NF-κB is constitutively active. DMF may also be useful as an adjuvant therapy to conventional anticancer drugs.

## 1. Introduction

Breast cancer is the most prevalent cancer in women worldwide. Breast cancer is classified as positive or negative for the estrogen receptor (ER), the progesterone receptor (PR), and human epidermal growth factor receptor 2 (HER2). The subtypes of breast cancer include luminal A (ER/PR-positive, Ki-67-low, and HER2-negative), luminal B (ER/PR-positive, Ki-67-high, and HER2-negative/positive), HER2-positive, and triple-negative breast cancer (TNBC; ER/PR/HER2-negative), which differ in terms of progression, recurrence rate, and treatment options [1]. TNBC accounts for about 20% of all breast cancers and has the highest three-year recurrence rate, resulting in a poor prognosis [2]. In recent years, the development of PARP-1 inhibitors and anti-PD-L1 antibodies has improved the response rate, but conventional chemotherapy, which has not significantly improved the prognosis, remains the basic treatment [3]. In addition, since TNBC is negative for ER, PR, and HER2, and because there is no clear therapeutic target molecule, a new treatment for TNBC is needed.

Recently, NF-κB p65 (RELA) gene expression was reported to be increased in TNBC compared to that in other subtypes, deeply involved in TNBC stem cell survival, and activated during the recurrence of breast cancers, including TNBC [4,5,6]. NF-κB, a transcription factor that promotes cell proliferation, differentiation, and survival, is involved in tumorigenesis and cancer survival in various solid tumors, including pancreatic, lung, cervical, prostate, breast, and gastric cancers [7]. Therefore, NF-κB may be a new therapeutic target molecule for TNBC.

Dimethyl fumarate (DMF) is a therapeutic agent for multiple sclerosis. The serum concentration of DMF is about 12 μM when administered at an oral dose of 240 mg/day in adults, and its safety has been confirmed [8]. In addition, DMF induces apoptosis and suppresses the growth of various cancer cells through the inhibition of NF-κB activation [9,10,11,12]. Therefore, DMF may be effective in TNBC, in which NF-κB is constitutively active.

In the present study, we examined the efficacy of DMF in TNBC cells with constitutive NF-κB activation and determined whether DMF has potential as an adjuvant to enhance the cytotoxic effects of paclitaxel and adriamycin in TNBC.

## 2. Results

### 2.1. Phosphorylated NF-κB p65 Protein and NFKBIA Gene Expression Is Associated with Prognosis in Patients with TNBC

We examined the association between the prognosis of patients with TNBC and the expression of phosphorylated NF-κB protein and NFKBIA, which encodes I-κBα, using the Kaplan–Meier Plotter. Increased expression of phosphorylated NF-κB p65 (Ser536) and decreased expression of NFKBIA were associated with poor overall survival (Figure 1A–D). In addition, the expression of RELA, which encodes NF-κB p65, in TNBC was higher than that in normal breast tissue, according to the analysis of the GSE dataset GSE45827 (Figure 1E). These findings indicate that NF-κB p65 activation is associated with poor prognosis in patients with TNBC.

### 2.2. DMF Induced Apoptosis in MDA-MB-231 and BT549 Cells

We investigated the activation and expression of NF-κB and I-κBα in MCF-10A, MDA-MB-231, and BT-549 cells. The activation and expression of NF-κB p65 were higher in MDA-MB-231 and BT-549 cells than in MCF-10A cells (Figure 2). In addition, the expression of I-κBα was lower in MDA-MB-231 and BT-549 cells than in MCF-10A cells (Figure 2). Next, we investigated the cytotoxic effects of DMF in MCF-10A, MDA-MB-231, and BT-549 cells. The cells were treated with 1–50 μM DMF, and the cell survival rate was examined. DMF markedly diminished the survival of MDA-MB-231 and BT-549 cells but did not affect the survival of MCF-10A cells (Figure 3A). In addition, DMF markedly increased the number of Annexin V–positive cells and the expression of cleaved caspase-3 in MDA-MB-231 and BT-549 cells (Figure 3B,C). These results suggest that DMF induces apoptosis in TNBC cells at concentrations that do not affect normal breast epithelial cells.

### 2.3. DMF Suppressed NF-κB p65 Nuclear Translocation and XIAP, Survivin, Bcl-xL, and Bcl-2 Expression in MDA-MB-231 and BT-549 Cells

We previously clarified that DMF inhibits the nuclear translocation of NF-κB p65 in tumor cells [9,12]. In this study, we examined whether DMF inhibits the nuclear translocation of NF-κB and the expression of other survival signals, such as JNK, p38MAPK, ERK1/2, and AKT, in MDA-MB-231 and BT-549 cells. DMF inhibited and enhanced the expression of nuclear and cytoplasmic NF-κB p65, respectively, in MDA-MB-231 and BT-549 cells but did not affect the activation of JNK, p38MAPK, ERK1/2, or AKT (Figure 4A).

NF-κB modulates the expression of apoptosis-regulated factors, such as Bcl-2 family proteins and IAP family proteins [13]. DMF inhibited the expression of Survivin, XIAP, Bcl-xL, and Bcl-2 without affecting Bim and Bax expression in MDA-MB-231 and BT-549 cells (Figure 4B). These results indicate that DMF-induced apoptosis occurs via the suppression of NF-κB p65 nuclear translation and the suppression of Survivin, XIAP, Bcl-xL, and Bcl-2 expression in TNBC cells.

### 2.4. DMF Enhanced the Cytotoxic Effects of Paclitaxel and Adriamycin in MDA-MB-231 and BT-549 Cells

Paclitaxel and adriamycin are used to treat patients with TNBC [14]. MDA-MB-231 and BT-549 cells were treated with DMF to examine whether the suppression of NF-κB enhances the cytotoxic effects of paclitaxel and adriamycin. The combined administration of DMF and paclitaxel or adriamycin markedly enhanced cell death compared with the administration of DMF, paclitaxel, or adriamycin alone (Figure 5A,B). In MCF10A cells, paclitaxel and adriamycin induced cell death, but DMF did not affect this result (Figure 5C). Moreover, in MDA-MB-231 cells, the combined administration of DMF and paclitaxel or adriamycin markedly enhanced the number of Annexin V–positive cells (Figure 6A). In addition, in MDA-MB-231 cells, the combined administration of DMF and paclitaxel or adriamycin significantly suppressed NF-κB nuclear translocation and Survivin, XIAP, Bcl-xL, and Bcl-2 expression, as well as elevated cleaved caspase-3 expression (Figure 6B).

## 3. Discussion

In this study, overexpression of phosphorylated NF-κB p65 and low expression of NFKBIA were correlated with poor prognosis in patients with TNBC. NF-κB p65 expression (nuclear and cytoplasmic) and I-κB expression were higher and lower, respectively, in MDA-MB-231 and BT-549 cells than in MCF-10A cells. The expression of RELA is higher in patients with TNBC than in patients with other breast cancer subtypes (luminal A, luminal B, and HER2-positive) and in normal breast tissue [4]. In addition, NFKBIA deletion is associated with poor disease-specific survival, recurrence-free survival, and distant metastasis survival in patients with TNBC [15]. Our findings indicated that the overexpression and activation of NF-κB p65 and the low expression of NFKBIA (I-κB) in TNBC were correlated with poor prognosis. Therefore, NF-κB p65 may be a therapeutic target molecule in TNBC.

At concentrations that were non-cytotoxic to the normal mammary cell line MCF-10A, DMF upregulated cleaved caspase-3 expression and increased the number of Annexin V–positive cells in MDA-MB-231 and BT-549 cells, indicating an apoptosis-enhancing effect. In addition, DMF suppressed NF-κB nuclear translocation but did not affect the activation of JNK, p38MAPK, ERK1/2, or AKT in MDA-MB-231 and BT-549 cells. DMF inhibits tumor growth and induces apoptosis in various tumor cells through the suppression of NF-κB activation [9,10,11,12]. DMF has been previously shown to be effective against malignant tumors in which NF-κB is constitutively activated. Therefore, DMF may be effective against TNBC, which exhibits overexpressed NF-κB.

NF-κB modulates the transcription and/or expression of various apoptosis-regulated factors, such as IAP family proteins and Bcl-2 family proteins [13]. In this study, DMF inhibited Survivin, XIAP, Bcl-xL, and Bcl-2 expression but did not affect Bax or Bim expression in BT-549 and MDA-MB-231 cells. Arsenic trioxide-induced apoptosis is associated with NF-κB suppression-mediated inhibition of Survivin, XIAP, cIAP2, and Bcl-xL expression in the acute promyelocytic leukemia cell line NB-4 [16]. Additionally, trichothecin-induced apoptosis downregulates the expression of Survivin, XIAP, Bcl-xL, Bcl-2, and cyclin D1 via abrogation of NF-κB activation in human cancer cells featuring constitutively active NF-κB [17]. Furthermore, methanol and chloroform extracts of Rumex dentatus induce apoptosis in MDA-MB-231 cells through the inhibition of NF-κB activation and suppress the expression of Survivin, XIAP, Bcl-xL, Bcl-2, and cyclin D1 [18]. Our results suggested that DMF-induced apoptosis was correlated with the downregulation of Survivin, XIAP, Bcl-xL, and Bcl-2 expression in TNBC cells.

In this study, DMF enhanced the apoptosis-inducing effect of paclitaxel and adriamycin via suppression of NF-κB activation and Survivin, XIAP, Bcl-xL, and Bcl-2 expression in MDA-MB-231 cells. In addition, DMF had no effect on the cytotoxicity of paclitaxel and adriamycin to MCF10A cells. DMF enhances the anti-melanoma efficacy of vemurafenib, a BRAF inhibitor, by suppressing multiple signaling pathways [19]. In a phase I clinical trial of DMF in combination with temozolomide and radiotherapy in patients with glioblastoma, a dose of 240 mg three times daily was reported to be safe and well tolerated by patients, with partial responses observed in 4 of 12 patients [20]. In addition, phase II clinical trials of DMF therapy have demonstrated efficacy and a high safety profile in patients with cutaneous T-cell lymphoma [21]. Since DMF has shown clinical efficacy and safety when used as monotherapy and in combination therapy for other types of cancer, DMF may be useful as an adjuvant to anticancer agents, such as paclitaxel and adriamycin, in TNBC.

DMF has been reported to suppress the constitutive and tumor necrosis factor α-induced activation of NF-κB and induce cell death in human breast cancer cells [22]. DMF has also been reported to inhibit MDA-MB-231 tumor growth in vivo [22]. Our present study clearly shows that DMF induces apoptosis through decreased Survivin, XIAP, Bcl-xL, and Bcl-2 expression via inhibition of NF-κB activation and potentiates the apoptosis-inducing effect of adriamycin and paclitaxel in human breast cancer cells. In addition, elevated phospho-NF-κB p65 (Ser536) and low NFKBIA (I-κB) expression were associated with poor prognosis in patients with TNBC; therefore, these expression states may be biomarkers for DMF therapy in TNBC. The limitation of our present study is that we were not able to confirm the enhanced antitumor effect of DMF in combination with adriamycin or paclitaxel in vivo. Therefore, these should be elucidated in the next study.

## 4. Materials and Methods

### 4.1. Cell Lines and Reagents

MCF-10A (normal breast epithelial cell line), MDA-MB-231 (TNBC cell line), and BT549 (TNBC cell line) were obtained from the American Tissue Culture Collection (Manassas, VA, USA). These cells were incubated with RPMI-1640 medium (Sigma, St Louis, MO, USA) containing 10% fetal bovine serum (Gibco, Carlsbad, CA, USA), 25 mM 4-(2-hydroxyethyl)-1-piperazine ethanesulfonic acid (FUJIFILM Wako), 100 U/mL streptomycin (Gibco), and 100 μg/mL penicillin (Gibco) in a 5% CO_2_ atmosphere. DMF, paclitaxel, and adriamycin were obtained from FUJIFLIM Wako (Tokyo, Japan). DMF and paclitaxel were dissolved in dimethyl sulfoxide (DMSO) and were diluted using phosphate-buffered saline (PBS; 0.05 M, pH 7.4). Adriamycin was dissolved in PBS.

### 4.2. Cell Survival and Apoptosis Detection Assay

Cell survival was measured by the trypan blue staining assay as previously described [23,24,25]. Briefly, cells were plated in 96-well plates at a concentration of 2 × 10^4^ cells/mL and were treated with various concentrations of DMF, paclitaxel, or adriamycin for 72 h. After incubation, cells were stained with 0.4% trypan blue solution, and dead and alive cells were counted. Apoptosis was detected using an Annexin V-FITC apoptosis detection kit (Nacalai Tesque, Inc., Kyoto, Japan) as previously described [26]. DMF-, paclitaxel-, or adriamycin-treated cells were washed three times in PBS and then resuspended in a binding buffer containing Annexin V-FITC. The cells were incubated for 15 min at room temperature and then analyzed using a BD-LSR Fortessa flow cytometer (Becton Dickinson, Bedford, MA, USA).

### 4.3. Immunoblotting

Nuclear and cytoplasmic fractions were collected using the ProteoExtract Subcellular Proteome Extraction Kit (Calbiochem, San Diego, CA, USA) and analyzed by immunoblotting assay using the following antibodies: anti-NF-κB p65 (#8242), anti-phospho-JNK (#9251), anti-JNK (#9252), anti-phospho-p38MAPK (#9211), anti-p38MAPK (#9212), phospho-ERK1/2 (#9101), anti-ERK1/2 (#9102), anti-phospho-Akt (#9271), anti-Akt (#9272), anti-XIAP (#2042), and anti-Survivin (#2803) (Cell Signaling Technology, Beverly, MA, USA); anti-Bcl-xL (sc-8392), anti-Bcl-2 (sc-7382), anti-Bax (sc-7480), anti-Bim (sc-374358), anti-caspase 3 (sc-7272), and anti-Lamin A/C (sc-7292) (Santa Cruz Biotechnologies, CA, USA); and anti-β-actin (A2228, Sigma, St Louis, MO, USA). The amount of detected protein was measured via densitometry using a CS analyzer (ATTO, Tokyo, Japan), and detected proteins were standardized to corresponding proteins.

### 4.4. Gene Expression Omnibus Dataset

The gene expression profiles in the microarray dataset (accession number GSE45827) were procured from the National Center of Biotechnology Information Gene Expression Omnibus database (http://www.ncbi.nlm.nih.gov/geo/ (accessed on 9 June 2022)). The expression of RELA in TNBC and normal breast tissues was analyzed.

### 4.5. Kaplan–Meier Plotter Analysis

The prognostic value of phospho-NF-κB p65 (Ser536) protein and NFKBIA expression in TNBC was calculated using the Kaplan–Meier Plotter (http://kmplot.com/analysis/ (accessed on 9 June 2022)), a database that unifies clinical data and protein expression data [27,28]. The patient samples were split into two groups (high and low expression) and compared using a Kaplan–Meier survival plot. The log-rank *p*-value was then calculated.

### 4.6. Statistical Analysis

Results are shown as the mean ± standard deviation (SD). Statistical analysis was executed using ANOVA with Dunnett’s test, and distinctions were considered significant at *p* < 0.05.

## 5. Conclusions

In conclusion, activation of NF-κB was associated with poor prognosis in patients with TNBC. In addition, DMF induced apoptosis by abrogating NF-κB activation and Survivin, XIAP, Bcl-xL, and Bcl-2 expression. DMF also enhanced the apoptosis-inducing effect of paclitaxel and adriamycin in TNBC, in which NF-κB is constitutively active. These findings suggest that DMF may be effective as a therapeutic agent against TNBC and as an adjuvant to conventional anticancer drugs.

## Figures and Tables

**Figure 1 ijms-23-08681-f001:**
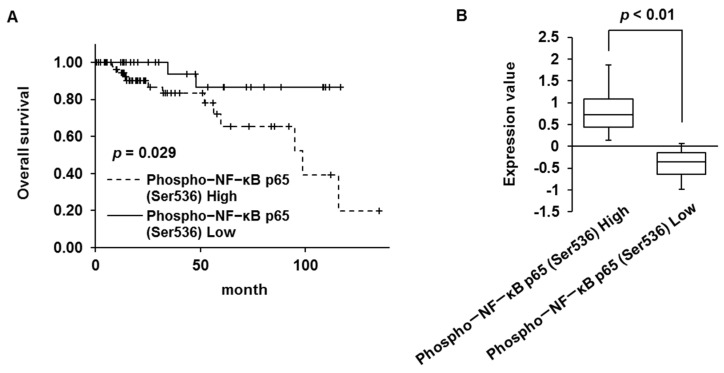
Activation of NF-κB p65 was associated with poor prognosis in patients with TNBC. (**A**) Relation between phospho-NF-κB p65 (Ser536) expression and overall survival. Survival rates were appraised using Kaplan−Meier Plotter. (**B**) High and low phospho-NF-κB p65 (Ser536) expression groups were analyzed using Kaplan−Meier Plotter. (**C**) Relation between NFKBIA expression and overall survival. Survival rates were appraised using Kaplan−Meier Plotter. (**D**) High and low NFKBIA expression groups were analyzed using Kaplan−Meier Plotter. (**E**) Expression of RELA in TNBC and normal tissues was analyzed using the GSE45827 dataset.

**Figure 2 ijms-23-08681-f002:**
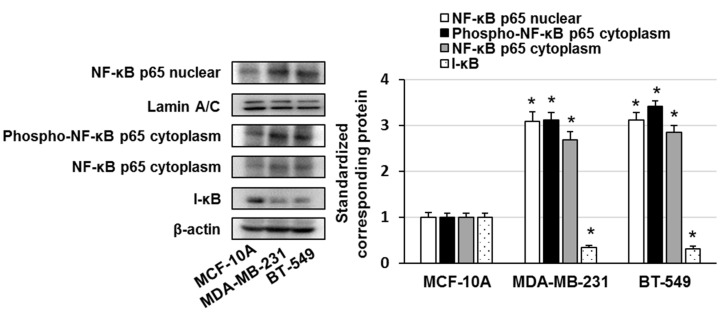
Cell lysates were evaluated by immunoblotting using the shown antibodies. The content of nuclear NF-κB p65, cytoplasmic phospho-NF-κB p65, cytoplasmic NF-κB p65, and I-κB was quantified and standardized to the content of Lamin A/C or β-actin. The results are representative of three independent experiments. * *p* < 0.01 vs. untreated cells.

**Figure 3 ijms-23-08681-f003:**
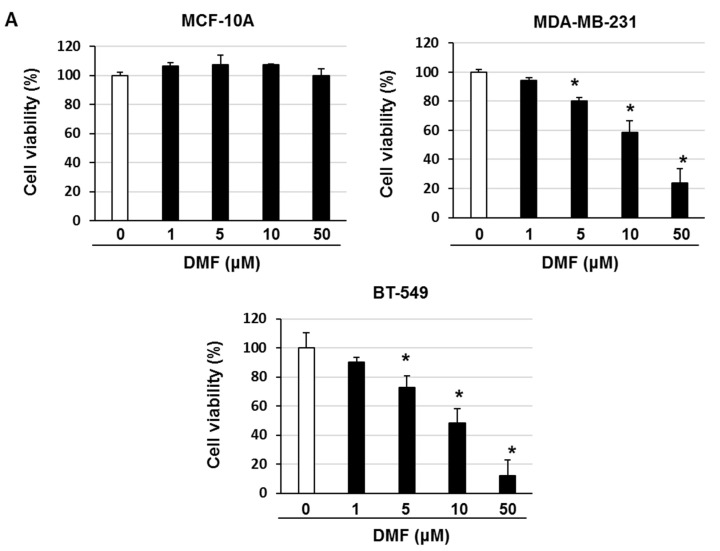
DMF induced apoptosis in MDA-MB-231 and BT-549 cells. (**A**) Viability of DMF-treated MCF-10A, MDA-MB-231, and BT-549 cells, as determined by the trypan blue staining assay. Cells were treated with the shown concentrations of DMF for 3 days. The results are representative of five independent experiments. * *p* < 0.01 vs. untreated cells. (**B**) MDA-MB-231 and BT-549 cells were treated with the shown concentrations of DMF for 72 h and then stained using an Annexin V apoptosis assay kit. The results are representative of four independent experiments. * *p* < 0.01 vs. untreated cells. (**C**) Cell lysates were evaluated by immunoblotting using the shown antibodies. The content of cleaved caspase-3 was quantified and standardized to the content of β-actin. The results are representative of three independent experiments. * *p* < 0.01 vs. untreated cells.

**Figure 4 ijms-23-08681-f004:**
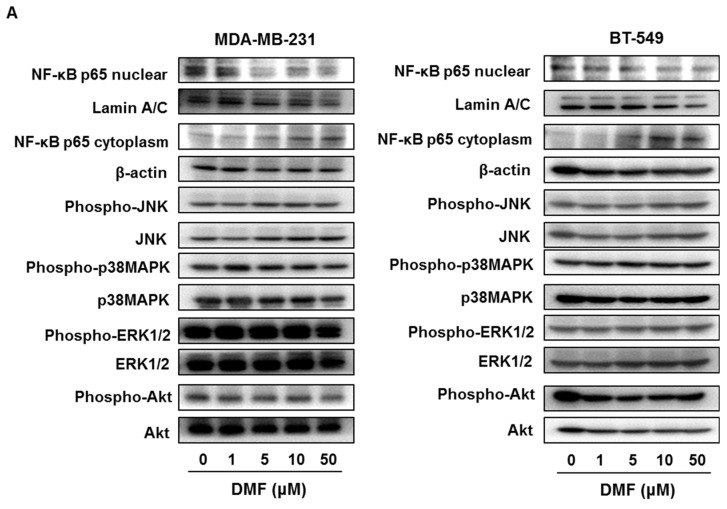
DMF suppressed NF-κB nuclear translocation and Survivin, XIAP, Bcl-xL, and Bcl-2 expression in MDA-MB-231 and BT-549 cells. (**A**) Cell lysates were evaluated by immunoblotting using the shown antibodies. The contents of nuclear NF-κB p65, cytoplasmic NF-κB p65, phospho-JNK, phospho-p38MAPK, phospho-ERK1/2, and phospho-Akt were quantified and standardized to the contents of Lamin A/C, β-actin, JNK, p38MAPK, ERK1/2, or AKT. The results are exemplary of three independent experiments. * *p* < 0.01 vs. untreated cells. (**B**) Cell lysates were evaluated by immunoblotting using the shown antibodies. The contents of Survivin, XIAP, Bcl-xL, Bcl-2, Bim, and Bax were quantified and standardized to the content of β-actin. The results are exemplary of three independent experiments. * *p* < 0.01 vs. untreated cells.

**Figure 5 ijms-23-08681-f005:**
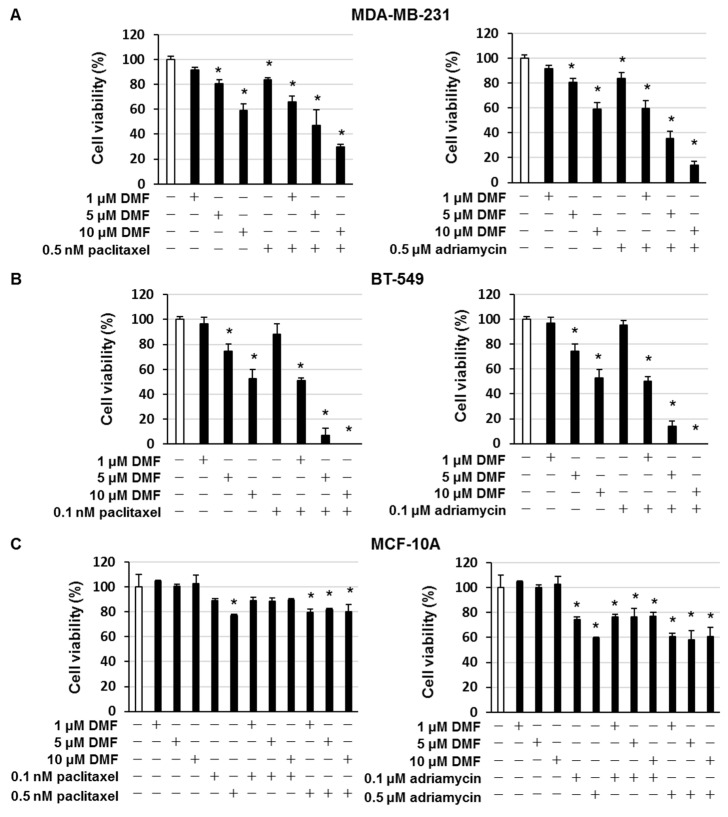
DMF enhanced the cytotoxic effect of paclitaxel and adriamycin in TNBC cells. (**A**) MDA-MB-231, (**B**) BT-549, and (**C**) MCF-10A cells were treated with the shown concentrations of DMF and paclitaxel or adriamycin. After incubation for 72 h, the number of dead cells was quantified using the trypan blue staining assay. The results are exemplary of five independent experiments. * *p* < 0.01 vs. untreated cells.

**Figure 6 ijms-23-08681-f006:**
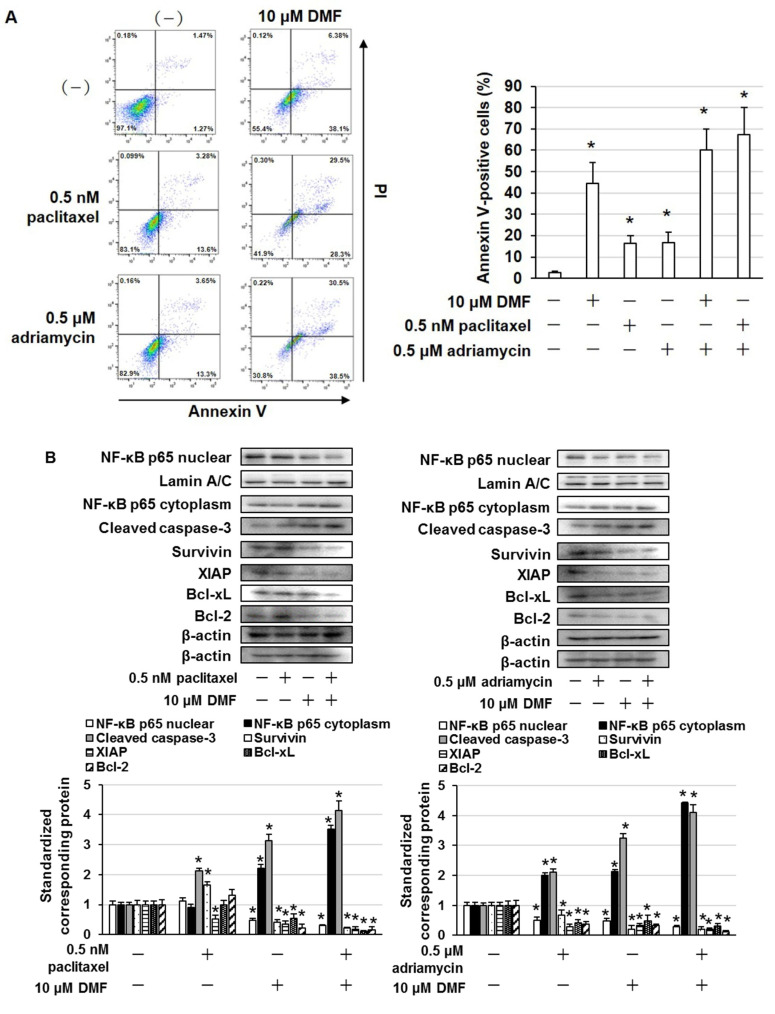
DMF enhanced the apoptosis-inducing effect of paclitaxel and adriamycin in MDA-MB-231 cells. (**A**) MDA-MB-231 cells were treated with the shown concentrations of DMF, paclitaxel, or adriamycin for 72 h and then stained using an Annexin V apoptosis assay kit. The results are exemplary of four independent experiments. * *p* < 0.01 vs. untreated cells. (**B**) DMF-, paclitaxel-, or adriamycin-treated cell lysates were evaluated by immunoblotting using the shown antibodies. The contents of nuclear NF-κB p65, cytoplasmic NF-κB p65, cleaved caspase-3, Survivin, XIAP, Bcl-xL, and Bcl-2 were quantified and standardized to the contents of Lamin A/C or β-actin. The results are exemplary of three independent experiments. * *p* < 0.01 vs. untreated cells.

## Data Availability

The data presented in this study are available on request from the corresponding author.

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
