# Peer review of "Dimethyl Fumarate Induces Apoptosis via Inhibition of NF-κB and Enhances the Effect of Paclitaxel and Adriamycin in Human TNBC Cells"

_ijms, 2022, doi:10.3390/ijms23158681_

Round 1

Reviewer 1 Report

The study by Tsurushima and colleagues is aimed at investigating the effect of Dimethyl fumarate in different cell lines of breast tumor, also in combination with paclitaxel and adriamycin. The results clearly show that the compound induces apoptosis in malignant cell lines by inhibiting Nf-kb signaling.  

Even if the results are interesting, they do not add much innovation to what already present in literature, please see the study by Kastrati et al. 2016 (Dimethyl Fumarate Inhibits the Nuclear Factor κB Pathway in Breast Cancer Cells by Covalent Modification of p65 Protein).

 This is the major point to be addressed by the Authors: what’s the innovation of their study in comparison the present literature?

Additional points to be addressed:

-       -  In general, the figures’ organization is very confusing. Excluding Fig. 1, the other ones present too many graphs all packed together. I suggest the Authors to re-organize them, in order to improve their quality and to help the reader in their consultation.  

-      -   Material and methods section is very poor. The Authors must better detail the cell culture conditions, as well as all the procedures.  

Reviewer 2 Report

In this manuscript, the authors show that activation of NF-kB p65 is implicated in triple-negative breast cancer (TNBC) prognosis. Furthermore, the experimental work in two TNBC cell lines supports the notion that dimethyl fumarate (DMF) induces apoptosis via inhibition of NF-kB and enhances the effects of Paclitaxel and Adriamycin. Because DMF is currently in clinical trials as a monotherapy and in combination with standard of care drugs, this work would enable in vivo studies that could lead to clinical trials in women with TNBC, which will be highly significant.

The laboratory work seems well done, and the conclusions are appropriate. The work should capture the interest of the wide TNBC audience, and it is timely. In this manuscript, the authors show that activation of NF-kB p65 is implicated in triple-negative breast cancer (TNBC) prognosis. Furthermore, the experimental work in two TNBC cell lines supports the notion that dimethyl fumarate (DMF) induces apoptosis via inhibition of NF-kB and enhances the effects of Paclitaxel and Adriamycin. Because DMF is currently in clinical trials as a monotherapy and in combination with standard of care drugs, this work would enable in vivo studies that could lead to clinical trials in women with TNBC, which will be highly significant.

The laboratory work seems well done, and the conclusions are appropriate. The work should capture the interest of the wide TNBC audience, and it is timely.

Round 2

Reviewer 1 Report

The Authors have amended the manuscript according to the suggestions.